# REINFORCEMENT-GUIDED SUBTASK DECOMPOSITION FOR UNIFIED VISION-LANGUAGE LEARNING

## ABSTRACT

Building truly versatile Vision-Language Models requires more than just scaling up training data and model size. While current models achieve impressive performance on their training tasks, they often fail when deployed on problems that require the same underlying skills but in different combinations—a limitation that hinders their adoption as general-purpose AI systems. This paper introduces a novel reinforcement learning framework that addresses this limitation by teaching a VLM to solve problems by composing a learned sequence of reusable, verifiable subtasks. Our key innovation is a reward function that guides the model to generate structured reasoning chains of these primitive subtasks through format-based verification, eliminating the need for detailed annotations of intermediate reasoning steps. This format-based reward provides a dense learning signal, enabling the model to master a flexible, procedural approach to problem-solving. Furthermore, these models can be flexibly transferred to spatial VQA tasks, demonstrating strong performance without any fine-tuning. This comprehensive cross-task transfer outperforms both standard supervised fine-tuned and reinforcement fine-tuned visual chain-of-thought baselines, while maintaining computational efficiency with only a 3B parameter model. Our findings show that learning to compose a sequence of fundamental vision skills is an effective and scalable strategy for building robust, general-purpose VLMs than learning monolithic, task-specific solutions.

## 1 INTRODUCTION

The pursuit of building universal, general-purpose Vision-Language Models (VLMs) has seen remarkable progress, largely driven by scaling up models and training data (Radford et al., 2021; Bai et al., 2023). A common paradigm involves fine-tuning these large pre-trained models on specific downstream tasks to achieve state-of-the-art performance (Liu et al., 2023). However, this task-specific fine-tuning approach faces fundamental limitations. It incurs prohibitive computational costs when adapting to multiple tasks, demands vast amounts of labeled data for each new application, and often leads to catastrophic forgetting, where improving performance on one task degrades capabilities on others (Kirkpatrick et al., 2017; Wang et al., 2023). Most critically, this approach produces models that excel only within their training domain but fail to transfer their learned capabilities to structurally different tasks, highlighting the persistent challenge of cross-task generalization in modern VLMs (Zhou et al., 2025).

We posit that the key to bridging this gap lies in learning to compose fundamental, reusable visual skills rather than task-specific solutions. Many complex vision-language tasks, though treated as distinct problems, can be decomposed into a sequence of common subtasks. For instance, both referring segmentation (e.g., "find the person wearing a red shirt") and object counting (e.g., "count all the vehicles in the parking lot") require fundamental capabilities like object detection and attribute recognition. If a model could explicitly master these primitive subtasks during fine-tuning on one task, it could potentially solve entirely different tasks by composing these skills in novel sequences (Zhao et al., 2024).

However, teaching a model to generate such decompositions is the central challenge, with two dominant paradigms presenting their own significant drawbacks. The most direct approach, supervised fine-tuning, would involve training the model on datasets containing explicit, step-by-step reason-

ing chains (Zhang et al., 2023). This method, while conceptually simple, is practically infeasible at scale due to the prohibitive cost of creating such detailed annotations (Fu et al., 2023). Furthermore, it forces the model to mimic human-defined reasoning paths, potentially limiting its ability to discover more effective or novel strategies (Rajaraman et al., 2020). The alternative paradigm, reinforcement learning (RL), circumvents the need for explicit labels by allowing the model to learn from a simple success signal based on the final answer (Pan et al., 2025; Liu et al., 2025a;b).

Yet, this approach faces challenges with sparse learning signals: when a long reasoning chain leads to a correct answer, the trajectory-level reward provides little guidance about which specific tokens or reasoning steps contributed to the success (Ouyang et al., 2024; Shao et al., 2024). This sparse and inefficient learning signal limits the model's ability to learn effective problem decomposition, highlighting a clear need for a method that can guide the model towards structured reasoning without the cost of full supervision or the ambiguity of sparse rewards.

To overcome these challenges, we propose a novel framework that teaches a VLM to decompose complex problems into a sequence of verifiable subtasks. Instead of relying on direct supervision for the decomposition itself, we pre-define a library of primitive subtasks and their expected output formats. The model is then trained using Group Relative Policy Optimization (GRPO) (Shao et al., 2024) to generate a reasoning chain of these subtasks. Our key innovation is a composite reward function that combines the final answer's accuracy with a dense, format-verification reward for correctly generating intermediate subtasks. This approach provides a stronger learning signal, guiding the model to learn subtask execution without explicit labels for the subtasks.

We demonstrate the efficacy of our method through extensive cross-task generalization experiments. Models trained solely on referring segmentation (RefCOCOg) task achieve strong zero-shot performance on counting benchmarks (CountBench, ReasonCount), while models trained on counting (PixMo-Point) task successfully generalize to complex segmentation tasks (ReasonSeg). Moreover, these models can be flexibly transferred to spatial VQA tasks (EgoOrientBench, V*Bench). Our framework, which leverages a dense, subtask-guided reward signal, significantly outperforms both supervised fine-tuning and a comparable reinforcement learning baseline that relies on a sparse, outcome-based reward. Remarkably, our 3B parameter model achieves competitive or superior performance compared to much larger specialized models, demonstrating that learning compositional skills is not only more generalizable but also more efficient. Our findings establish that teaching models to compose a sequence of subtasks represents a more effective and scalable path toward truly general-purpose vision-language understanding.

## 2 RELATED WORKS

Our work addresses the fundamental challenge of cross-task generalization in vision-language models through compositional learning. We review relevant work in three key areas: approaches to generalizable VLMs, task decomposition methods, and reinforcement learning for reasoning.

**Cross-task generalization in VLMs.** The dominant paradigm for building vision-language models involves massive-scale pre-training followed by task-specific fine-tuning (Radford et al., 2021; Bai et al., 2023; Liu et al., 2023). While this produces strong in-domain performance, these models often fail to transfer learned capabilities across task boundaries—a limitation we term the cross-task generalization gap. Previous attempts to address this limitation include multi-task learning (Caruana, 1997), where models are trained simultaneously on multiple tasks, and meta-learning approaches (Finn et al., 2017) that aim to learn task-agnostic representations. However, these methods typically require access to multiple tasks during training and still struggle with truly novel task structures. Recent work on prompt-based learning (Zhou et al., 2022) and adapter modules (Houlsby et al., 2019) has shown promise for efficient task adaptation but does not fundamentally solve the problem of zero-shot transfer to structurally different tasks. Our approach differs fundamentally by learning common subtasks that can be recombined for novel tasks, enabling genuine cross-task transfer without task-specific adaptation.

**Compositional reasoning and task decomposition.** The idea that complex reasoning emerges from composing simpler operations has deep roots in cognitive science and has inspired various computational approaches. In the vision-language domain, neural module networks (Andreas et al., 2016) pioneered the idea of composing specialized modules for different subtasks. More recently,

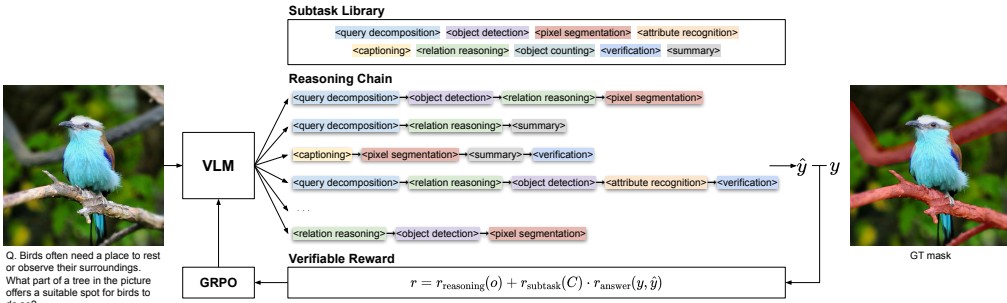

Figure 1: Architecture of the proposed method for cross-task generalization. The model takes a visual input and a textual query, decomposing the task into subtasks using a library of predefined skills. The model learns to generate structured reasoning chains. These learned compositional skills enable zero-shot generalization to unseen tasks by recomposing the same primitive subtasks in different sequences. The reward function verifies (1) the final answer accuracy, (2) the structural format, and (3) the validity of subtask outputs, guiding the model to learn generalizable reasoning strategies.

Chain-of-Thought (CoT) methods (Wei et al., 2022) have shown that explicit step-by-step reasoning improves performance on complex tasks, and further extensions such as Visual CoT (Zhang et al., 2023) have demonstrated the applicability of this approach to multimodal inputs. Another line of work that leverages multimodal inputs for task decomposition is ViperGPT (Surís et al., 2023), which proposes decomposing vision tasks into executable programs. However, these methods typically require either explicit supervision for decomposition strategies or rely on large language models' existing reasoning capabilities, limiting their applicability to vision-centric tasks. Nowadays, models such as Chain-of-Spot (Liu et al., 2024b) and FAST (Sun et al., 2024) have emerged, enabling Vision-Language Models (VLMs) to process visual and textual inputs jointly through a unified Chain-of-Thought (CoT) framework. However, these approaches still suffer from the drawback of requiring massive amounts of training data to both learn the target tasks and maintain performance on existing VQA benchmarks. Our method learns to discover and compose primitive subtasks through reinforcement learning, requiring neither explicit decomposition labels nor pre-existing reasoning capabilities.

**Reinforcement learning for VLMs.** Reinforcement learning offers a promising alternative to supervised fine-tuning by allowing models to learn from reward signals rather than explicit labels. Recent advances in RL for large models, including RLHF (Christiano et al., 2017), PPO adaptations (Schulman et al., 2017), and simplified methods like DPO (Rafailov et al., 2023) and GRPO (Shao et al., 2024), have made RL training more stable and efficient. Many works (Liu et al., 2025a;b) have leveraged the recently proposed GRPO to endow models with new reasoning capabilities. However, one of the key limitations of these RL methods is the sparsity of reward signals (Chen et al., 2025), which makes it difficult for models to effectively learn complex, multi-step reasoning procedures. Our contribution is a novel reward design that provides dense, format-based feedback on intermediate reasoning steps, enabling the model to learn verifiable subtasks without explicit supervision.

**Proposed method.** Our approach uniquely combines three key insights: (1) complex vision-language tasks share common underlying subtasks that can be explicitly identified and learned, (2) format-based verification provides a dense and effective reward signal for learning these subtasks through RL without expensive supervision, and (3) subtask learning enables genuine cross-task generalization rather than task-specific adaptation. This represents a fundamental shift from the prevailing paradigm of task-specific fine-tuning toward a more flexible, efficient approach to building general-purpose vision-language models.

## 3 METHODOLOGY

We propose a novel framework that equips VLMs with the ability to decompose complex problems into a collection of verifiable subtasks, enabling them to generalize to arbitrary tasks. To demonstrate this capability, we focus on two structurally different vision tasks: referring segmentation and object

Table 1: Pre-defined subtasks and their output formats. The model is trained to generate answers for these subtasks, which are then used to solve the main task. Each subtask is designed to capture a specific aspect of the overall task, allowing for modular and flexible problem-solving.

| Subtask | Description | Output Format |
|---|---|---|
| Query Decomposition | Decomposes the main query into smaller, manageable sub-queries. | `{"main_query": str, "sub_queries":[str, ...]}` |
| Object Detection | Detects all significant objects in the image and assigns them a unique ID. | `[{"id": int, "class": str, "bbox_2d":[x1,y1,x2,y2]}]` |
| Pixel Segmentation | Segments objects and their precise segmentation points. | `[{"id": int, "bbox_2d": [x1,y1,x2,y2], "point_2d":[x,y]}]` |
| Attribute Recognition | Identifies specific attributes like color, state, or material for a detected object. | `[{"id": int, "attributes":[str,...]}]` |
| Captioning | Makes a caption for the image based on the detected objects. | `[{"caption": str, "target_ids": [int,...]}]` |
| Relation Reasoning | Finds spatial or semantic relationships between two objects, e.g., "object_1 is on top of object_2". | `[{"object_1":int, "relation":str, "object_2":int}]` |
| Object Counting | Counts the number of objects of a specific class. | `[{"class": str, "count": int, "target_ids": [int,...]}]` |
| Verification | Verifies a previous step's result and determines its validity. | `{"claim": str, "verification": str, "status":"OK\|FAIL"}` |
| Summary | Summarizes the reasoning process. | `{"summary": str}` |

counting. For this, we identified common subtasks underlying various vision-language problems and fine-tuned a model using Group Relative Policy Optimization (GRPO).

As illustrated in Figure 1, for a given image $I$ and a textual query $Q$, the model generates the output $o$. The output consists of a reasoning chain $C = \{s_1, s_2, ..., s_N\}$ with $N$ subtask outputs, followed by a final solution $\hat{y}$. The entire process is trained end-to-end by maximizing the total reward for its generated outputs. First, we briefly introduce the GRPO training framework in Section 3.1. We then detail our subtask decomposition approach in Section 3.2. Finally, we present the verifiable reward function that enables the model to learn this decomposition in Section 3.3.

### 3.1 GRPO TRAINING FRAMEWORK

We employ Group Relative Policy Optimization (GRPO) (Shao et al., 2024) to fine-tune our policy model $\pi_\theta$. For each input $(I, Q)$, GRPO samples a group of $G$ outputs $\{o_1, ..., o_G\}$ from the policy. The reward function, which we will detail later, provides a scalar reward $r_i$ for each output $o_i$. The core of GRPO lies in its advantage estimation. Instead of a learned value function, it uses the statistics of the sampled group to normalize the rewards as

$$\hat{r}_i = \frac{r_i - \text{mean}(\{r_j\}_{j=1}^G)}{\text{std}(\{r_j\}_{j=1}^G)}. \tag{1}$$

From this outcome-level supervision, the advantage $\hat{A}_{i,t}$ at token $t$ is constant as reward $\hat{r}_i$, as the entire output $o_i$ receives the same reward signal. The policy is then updated by maximizing a clipped surrogate objective, similar to PPO, using this advantage. The full objective function, which includes a KL divergence penalty against a reference policy ($\pi_{\text{ref}}$) to prevent over-optimization, follows the standard formulation presented by Shao et al. (2024). As we do not perform task-specific supervised fine-tuning, we set $\pi_{\text{ref}}$ to be the initial policy $\pi_{\theta_{\text{init}}}$ from the weights of the pre-trained model.

### 3.2 STRUCTURED SUBTASK DECOMPOSITION

Our central idea is to guide the model to learn a compositional procedure without explicit supervision for the decomposition itself. We achieve this by training the model to generate a reasoning chain $C$ as a sequence of structured, verifiable subtask outputs, which are then used to produce a final solution $\hat{y}$. Given an image $I$ and query $Q$, the model generates the reasoning chain by sequentially producing outputs for a series of subtasks $s_i$. The generation process is factorized as

$$p(\hat{y}, C|Q, I) = p(\hat{y}|Q, I, C) \prod_{i=1}^{N} p(s_i|Q, I, s_{1:i-1}), \tag{2}$$

where each $s_i$ is a textual output corresponding to a subtask from a predefined library of subtasks.

Table 1 details all subtasks in our library with their output formats. Our library encompasses fundamental subtasks underlying diverse vision-language tasks, each with a strict JSON schema enabling automatic validation without manual annotation of reasoning chains. This compositional approach enables cross-task transfer: models learn to flexibly recombine validated subtasks rather than memorizing task-specific solutions, allowing knowledge acquired on one task to transfer naturally to problems with different surface structures but shared underlying operations.

### 3.3 VERIFIABLE REWARD FUNCTION

The model learns to produce meaningful decompositions through a composite reward function that provides dense feedback on both structural correctness and the answer accuracy. For a generated output $o = (C, \hat{y})$ and the ground-truth solution $y$, the reward is defined as:

$$r = r_{\text{reasoning}}(o) + r_{\text{subtask}}(C) \cdot r_{\text{answer}}(y, \hat{y}). \tag{3}$$

**Reasoning format reward.** The reasoning format reward $r_{\text{reasoning}}$ ensures proper output structure through binary verification: it checks whether the final answer follows the required task-specific format within designated answer tags, and whether the reasoning process contains properly structured subtask executions within thinking tags. This provides immediate structural feedback without requiring semantic evaluation.

**Subtask format reward.** The subtask reward $r_{\text{subtask}}$ provides dense feedback on the quality and diversity of intermediate reasoning steps:

$$r_{\text{subtask}} = \underbrace{\left(\frac{N_{\text{valid}}}{N}\right)}_{\text{Base Accuracy}} \times \underbrace{\left(1 - \alpha^{N_{\text{valid}}}\right)}_{\text{Exploration Bonus}} \times \underbrace{\left(\beta^{N_{\text{repeated}}}\right)}_{\text{Repetition Penalty}}, \tag{4}$$

where $N_{\text{valid}}$ is the number of correctly formatted subtasks, and $N_{\text{repeated}}$ is the number of subsequently repeated subtask instances. The base accuracy term ensures that only properly formatted subtasks contribute to the reward. The exploration bonus term $(1 - \alpha^{N_{\text{valid}}})$ creates an exponential reward structure that increasingly benefits models using more valid subtasks. This exponential structure provides diminishing returns as more subtasks are used, preventing the model from generating arbitrarily long reasoning chains while still incentivizing meaningful task decomposition. Specifically, the reward increases steeply for the first few valid subtasks but saturates as $N_{\text{valid}}$ grows larger, striking a balance between thorough task decomposition and concise reasoning. The repetition penalty $\beta^{N_{\text{repeated}}}$ discounts excessive reuse of identical subtasks. Together, these components promote reasoning chains that are both structurally correct and strategically diverse, enabling the model to develop flexible problem-solving approaches rather than rigid, repetitive patterns. We choose $\alpha = 0.5$ and $\beta = 0.9$ based to balance reward signal strength and reasoning diversity.

**Answer accuracy reward.** The accuracy reward $r_{\text{answer}}$ grounds the model's learning in actual problem-solving performance. This component varies by IoU for the segmentation task, and counting accuracy with localization precision for the counting task. The detail of the reward calculation for each task is provided in Appendix A.2. By multiplying the subtask reward with task performance, we ensure that diverse, well-structured reasoning is rewarded only when it leads to correct solutions.

This composite reward design is crucial as it creates a balanced learning signal. It simultaneously pushes the model to generate a correct final answer while also guiding it to produce a well-structured and verifiable reasoning process. This prevents the model from finding "shortcut" solutions and instead encourages the learning of a robust, generalizable problem-solving procedure.

## 4 EXPERIMENTS

We conduct comprehensive experiments to validate our framework's in-domain and cross-task generalization capabilities. We train models on single vision-language tasks and evaluate their ability to transfer learned compositional skills to structurally different domains, comparing our approach against standard fine-tuning and unstructured reasoning baselines.

## 4.1 EXPERIMENTAL SETUP

### 4.1.1 TRAINING TASKS

We select two training tasks that require fundamentally different reasoning approaches yet share core compositional subtasks. RefCOCOg (segmentation) (Yu et al., 2016) requires models to decompose complex referring expressions and generate precise spatial masks through bounding box and point predictions. This task encourages learning subtasks like object detection, attribute recognition, and spatial relationship reasoning. To demonstrate the efficiency of our compositional approach, we use only 1,000 samples from the training set—a fraction of typical fine-tuning requirements—while achieving competitive performance through effective skill transfer. PixMo-Point (counting by localization) (Deitke et al., 2025) presents a dual challenge where models must both enumerate objects and predict their precise 2D coordinates for each instance. Additionally, we require models to predict the exact count of target objects to ensure precise quantitative reasoning. This task naturally requires diverse subtasks to solve. We use 1,000 samples containing fewer than 10 target objects for training.

### 4.1.2 EVALUATION BENCHMARKS

We design our evaluation to test the model's ability to learn transferable compositional subtasks by assessing performance across three distinct domains that require different combinations of fundamental skills.

Segmentation tasks test fine-grained spatial reasoning and compositional language understanding. We evaluate on ReasonSeg (Lai et al., 2024), which demands complex logical reasoning for mask generation. This benchmark evaluates whether counting-trained models can transfer learned subtasks like object detection and attribute recognition to precise localization tasks.

Counting tasks assess quantitative reasoning and global scene understanding. We use CountBench (Paiss et al., 2023) for basic counting capabilities, and extend our evaluation with a custom benchmark called ReasonCount to probe more complex compositional skills. ReasonCount comprises two challenging tasks: (1) referring expression counting that integrates attribute recognition with spatial reasoning, and (2) multi-object arithmetic requiring object categorization and numerical computation. Built on PixMo-Point (Deitke et al., 2025), this benchmark determines whether segmentation-trained models can repurpose their subtasks for numerical reasoning.

Spatial VQA tasks evaluate spatial understanding and visual search skills. We assess performance on EgoOrientBench (Jung et al., 2024) for egocentric orientation understanding and V*Bench (Wu & Xie, 2024) for guided visual search in high-resolution images. These benchmarks reveal whether both training regimes produce generalizable subtasks that transfer beyond their original contexts. We evaluate on 500 randomly sampled examples from EgoOrientBench's most challenging "choose" task and the full V*Bench validation set.

This evaluation design tests our core hypothesis: models learning compositional subtasks should successfully recombine these skills across structurally different domains, with success indicating genuine skill transfer rather than superficial task similarities.

### 4.1.3 BASELINES

We compare our method against strong baselines to see the effectiveness of our learning approach.

Qwen2.5-VL-3B serves as our zero-shot baseline, representing the base model's inherent capabilities across multiple vision-language tasks without any task-specific fine-tuning.

SFT (supervised fine-tuning) is the standard approach, where models are trained to directly predict final answers without intermediate reasoning steps, serving as a conventional fine-tuning baseline.

VCoT (visual chain-of-thought with RL) provides the most direct comparison to our approach, as it uses the same GRPO training framework but with unstructured natural language reasoning instead of structured subtasks. Like our method, VCoT (RL) generates reasoning within `<think>` tags before producing answers in `<answer>` tags, but its reward function consists only of format verification and final accuracy rewards, lacking our subtask format reward component. This baseline directly

Table 2: Comprehensive cross-task generalization results. Models are trained on either RefCOCOg (segmentation) or PixMo-Point (counting) and evaluated across segmentation, counting, and spatial VQA benchmarks to assess both in-domain and cross-task performance. Green and red numbers indicate the increase or decrease compared to the Qwen2.5-VL-3B baseline.

| Method | Segmentation | | Counting | | Spatial VQA | |
|---|---|---|---|---|---|---|
| | ReasonSeg-val (GIoU) | ReasonSeg-test (GIoU) | CountBench (Acc) | ReasonCount (Acc) | EgoOrientBench (Choice-Acc) | V* (Acc) |
| *Zero-shot Baseline* | | | | | | |
| Qwen2.5-VL-3B | 54.5 | 47.6 | 72.5 | 14.0 | 28.2 | 70.7 |
| *Trained on RefCOCOg (Segmentation)* | | | | | | |
| SFT | 55.3 (+0.8) | 47.9 (+0.3) | 72.3 (-0.2) | 14.0 (+0.0) | 28.0 (-0.2) | 72.8 (+2.1) |
| VCoT (RL) | 58.6 (+4.1) | **57.3** (+9.7) | 67.8 (-4.7) | **15.0** (+1.0) | 36.4 (+8.2) | 70.2 (-0.5) |
| **Ours** | **59.4** (+4.9) | 57.1 (+9.5) | **73.3** (+0.8) | 14.8 (+0.8) | **36.8** (+8.6) | **72.8** (+2.1) |
| *Trained on PixMo-Point (Counting)* | | | | | | |
| SFT | 55.3 (+0.8) | 47.6 (+0.0) | 72.1 (-0.4) | 14.0 (+0.0) | 27.8 (-0.4) | 72.8 (+2.1) |
| VCoT (RL) | 35.9 (-18.6) | 36.6 (-11.0) | **77.8** (+5.3) | 20.2 (+6.2) | 32.6 (+4.4) | 72.2 (+1.5) |
| **Ours** | **56.9** (+2.4) | **55.4** (+7.8) | 76.6 (+4.1) | **20.5** (+6.5) | **37.8** (+9.6) | **73.3** (+2.6) |

isolates the contribution of structured subtask decomposition by controlling for the reinforcement learning framework.

### 4.1.4 Implementation details

We use Qwen2.5-VL-3B (Bai et al., 2025) as the base model, with a frozen SAM2-Large (Ravi et al., 2024) as the segmentation mask decoder for segmentation tasks. Training is performed with GRPO using a batch size of 16 and the number of samples $G = 8$ per input. The learning rate is set to $10^{-6}$ with weight decay of 0.01. We use $\alpha = 0.5$ for exploration bonus and $\beta = 0.9$ for repetition penalty term. More details are in the Section A.1 of the Appendix.

## 4.2 Main Results

**In-domain and cross-task generalization.** Table 2 summarizes the in-domain and cross-task generalization results. Our method consistently demonstrates strong and stable performance across all evaluated benchmarks, maintaining high score on both segmentation and counting tasks while effectively transferring knowledge across domains. In contrast, SFT and VCoT (RL) often show substantial drops when applied to tasks they were not trained on, and only occasionally achieve comparable results. For example, when trained on PixMo-Point, VCoT (RL) performs well on counting benchmarks but suffers a sharp decline on segmentation tasks. By comparison, our approach not only preserves in-domain performance but also consistently improves cross-task generalization. This demonstrates that the learned compositional reasoning chains with reusable subtasks provide a flexible procedural strategy that can be effectively applied across diverse tasks.

A similar pattern emerges in spatial VQA benchmarks, where our method achieves the highest gains across all tasks. While SFT and VCoT (RL) show inconsistent transfer, our method achieves steady gains on both EgoOrientBench and V*Bench. This suggests that the compositional subtasks learned by our approach generalize beyond segmentation and counting to support complex relational reasoning and guided visual search, highlighting the broader applicability of our framework to diverse visual-linguistic challenges.

Taken together, these findings underscore the key advantage of our method: rather than over-optimizing for a single task, it promotes stable and transferable learning of reusable subtasks. Consequently, our method offers a more reliable pathway to cross-task generalization than existing SFT or VCoT (RL) baselines.

**Comparison with specialist models.** Table 5 compares our 3B parameter model against larger specialist models on segmentation, counting, and spatial VQA tasks. Task-specific methods often rely on training massive parameter models with large-scale, task-specific data, achieving strong results within their target domain but leaving their effectiveness on unseen tasks during finetuning uncertain. In contrast, our proposed method, trained on only 1k samples, not only secures high in-domain performance but also generalizes effectively to other different tasks. This demonstrates the

Table 3: Ablation study on exploration bonus and repetition penalty in $r_{\text{subtask}}$. Both factors play a crucial role for learning appropriate compositional skills.

| Reward Components | | ReasonSeg (GIoU) | |
| --- | --- | --- | --- |
| Exploration Bonus | Repetition Penalty | val | test |
| | | 47.9 | 44.5 |
| ✓ | | 58.2 | 54.0 |
| ✓ | ✓ | **59.5** | **57.1** |

Table 4: Ablation study on the hyperparameter $\beta$ in repetition penalty. The model is trained on RefCOCOg and evaluated on ReasonSeg and CountBench.

| $\beta$ | ReasonSeg (IoU) | | CountBench |
| --- | --- | --- | --- |
| | val | test | Accuracy |
| 0.5 | 56.9 | 55.3 | 68.2 |
| 0.9 | **59.5** | **57.1** | **70.5** |
| 1.0 | 58.2 | 54.0 | 55.8 |

Table 5: Comparison with specialist models. Our 3B model trained on 1k samples achieves competitive performance with larger task-specific models. $^{\dagger}$ indicates the evaluation was conducted using 500 subsets of EgoOrientBench. $^{\ddagger}$ means the results obtained by finetuning on full EgoOrientBench.

| Model | Segmentation (GIoU) | | Counting (Acc) | | Spatial VQA (Acc) | |
| --- | --- | --- | --- | --- | --- | --- |
| | ReasonSeg-val | ReasonSeg-test | CountBench | ReasonCount | EgoOrientBench | V* |
| *Segmentation Specialists* | | | | | | |
| OV-Seg (Liang et al., 2023) | 28.5 | 26.1 | - | - | - | - |
| SEEM (Zou et al., 2023) | 25.5 | 24.3 | - | - | - | - |
| Grounded-SAM (Ren et al., 2024) | 26.0 | 21.3 | - | - | - | - |
| LISA-13B (Lai et al., 2024) | 56.2 | 51.7 | - | - | - | - |
| Seg-Zero-3B (Liu et al., 2025a) | 62.6 | 56.1 | - | - | - | - |
| Seg-Zero-7B (Liu et al., 2025a) | **62.6** | **57.5** | - | - | - | - |
| *Counting Specialists* | | | | | | |
| CLIP-B/32 (Paiss et al., 2023) | - | - | 31.7 | - | - | - |
| CLIP-Count (Paiss et al., 2023) | - | - | 75.9 | - | - | - |
| *Spatial VQA Specialists* | | | | | | |
| mPLUG-Owl2-7B (Jung et al., 2024) | - | - | - | - | 28.5 | - |
| InternVL2-4B (Jung et al., 2024) | - | - | - | - | 31.4 | - |
| LLaVA-1.5-7B (Liu et al., 2024a) | - | - | - | - | 33.7$^{\ddagger}$ | 48.7 |
| SEAL (Wu & Xie, 2024) | - | - | - | - | 33.7 | **75.4** |
| *Multi-Modal LLMs* | | | | | | |
| Qwen2.5-VL-3B (Bai et al., 2025) | 54.5 | 47.6 | 72.5 | 14.0 | 28.2$^{\dagger}$ | 70.7 |
| Qwen2.5-VL-7B (Bai et al., 2025) | 56.9 | 52.1 | 76.0 | 17.5 | 37.6$^{\dagger}$ | 70.2 |
| *Our Method (3B)* | | | | | | |
| Trained on RefCOCOg | 59.4 | 57.1 | 73.2 | 14.8 | 36.8$^{\dagger}$ | 72.8 |
| Trained on PixMo-Point | 56.9 | 55.4 | **76.6** | **20.5** | **37.8**$^{\dagger}$ | 73.3 |

efficiency of compositional learning: by leveraging subtask decomposition, our approach achieves robust generalization without sacrificing in-domain accuracy, enabling superior performance across both familiar and other different tasks.

## 4.3 ABLATION STUDIES

**Reward function components.** Table 3 analyzes the contribution of different reward components: exploration bonus and repetition penalty in Eq 4. We trained the models on RefCOCOg and evaluate them on ReasonSeg and CountBench to analyze the impact of different components. Without either component, the model shows considerably low performance on ReasonSeg. Adding the exploration bonus leads to a substantial improvement, and further incorporating the repetition penalty achieves the best performance on ReasonSeg. This demonstrates that both components are crucial for learning compositional skills.

**Ablation study on $\beta$** We further investigate the impact of the repetition penalty weight $\beta$ in the subtask reward formulation (Eq. 4). As shown in Table 4, we experiment with different values of $\beta$ to understand its influence on performance. When $\beta = 1.0$, the repetition penalty completely ignores the repetition of subtasks, which may lead to the underutilization of essential skills and suboptimal performance on ReasonSeg. At $\beta = 0.5$, the model shows relatively good performance but still falls short of the results at $\beta = 0.9$. This suggests that moderately discouraging subtask repetition is beneficial for learning the robust and diverse skill set necessary for effective reasoning.

## 4.4 SUBTASK ANALYSIS

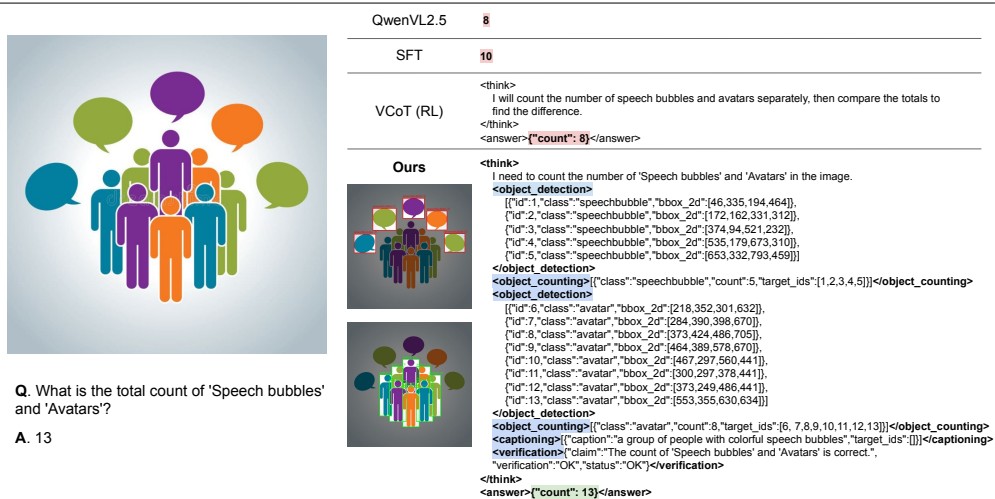

Figure 2: Example reasoning outputs on a ReasonCount arithmetic task requiring addition of speech bubbles and avatars (correct answer: 13). Our method generates structured subtask sequences, while baselines produce unstructured text with incorrect results.

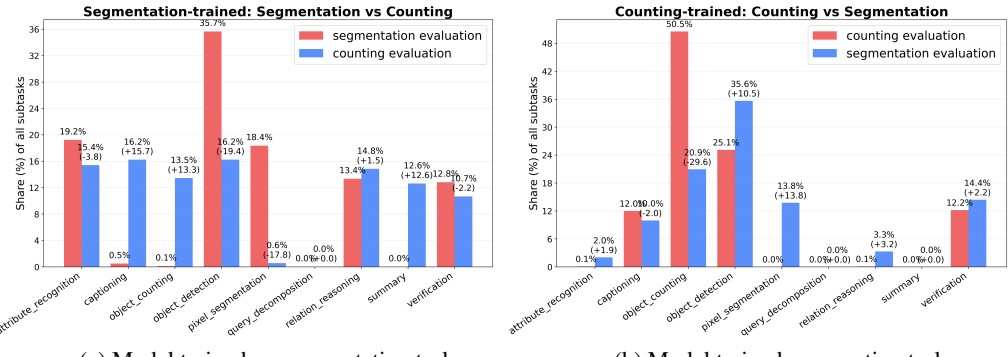

(a) Model trained on segmentation task.          (b) Model trained on counting task.

Figure 3: Task-specific subtask composition in reasoning. We compare subtask usage distributions (normalized by total subtasks) when evaluating the same model on different tasks. Distribution shifts indicate the model adapts its subtask selection to the current task effectively.

**Qualitative analysis of reasoning chains.** Figure 2 illustrates the effectiveness of structured sub-task decomposition for complex reasoning tasks. Our method systematically breaks down the multi-step arithmetic problem using a logical sequence of subtasks, <object_detection> followed by <object_counting>, leading to the correct answer through verifiable intermediate steps. This structured approach provides interpretable insights into the model's reasoning process, allowing for better understanding of each step. In contrast, baseline methods produce incorrect counts or irrelevant reasoning, lacking reliable performance on complex tasks. The clear subtask structure demonstrates how our framework enables both improved accuracy and enhanced interpretability compared to unstructured reasoning approaches.

**Analysis on the usage of subtasks.** Figure 3 reveals how the model makes logical, task-driven adjustments to its reasoning process. For instance, the segmentation-trained model (Figure 3a) adjusts its compositional pattern to object_counting and captioning which are essential for enumeration and scene understanding. Conversely, the model trained on the counting task (Figure 3b) shows a heightened reliance on pixel_segmentation and object_detection to tackle segmentation, deploying skills crucial for precise spatial localization. These purposeful shifts in subtask usage confirm that the model is not executing a memorized procedure, but is instead dynamically composing skills in response to specific task demands, enables the effective cross-task transfer observed in our results.

## 5 CONCLUSION

This work shows that teaching Vision-Language Models to break down complex tasks into reusable subtasks leads to much better generalization. Our key contribution is a reinforcement learning approach that uses format-based rewards to help models learn a set of verifiable subtasks without requiring expensive manual labels for task decomposition. This provides dense feedback that addresses the sparse reward problem common in learning multi-step procedures. We validate our method by demonstrating strong zero-shot performance when transferring from training on just one task to new, different tasks. This generalization comes without hurting performance on existing benchmarks—a common problem with standard fine-tuning approaches. Our results suggest that learning to combine subtasks is more effective than memorizing specific solutions for building versatile model.

ACKNOWLEDGMENTS

This work was supported by 42dot research project. We thank 42dot for their support and collaboration in this research.

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

## A SUPPLEMENTARY MATERIAL

### A.1 TRAINING CONFIGURATION DETAILS

**Training Settings.** Table 6 provides complete hyperparameter configurations used in our experiments. We use VeRL (Sheng et al., 2025) as the base codebase for implementing GRPO training. The RL training is conducted on 8 NVIDIA A6000 48GB GPUs, with a batch size of 16 and a GRPO group size of 8. The maximum sequence length is set to 2000 tokens. We employ a constant learning rate of 1e-6 with an AdamW optimizer, and train for 250 steps. We apply the KL divergence term with a coefficient of 1e-2 to stabilize training.

Table 6: Complete hyperparameter settings for all experiments.

| Hyperparameter | Value |
|---|---|
| Base Model | Qwen2.5-VL-3B |
| Optimizer | AdamW |
| Learning Rate | 1e-6 |
| Weight Decay | 0.01 |
| KL Coefficient | 1e-2 |
| Batch Size | 16 |
| GRPO Samples per Input | 8 |
| Max Sequence Length | 2000 |
| Total Training Steps | 250 |
| *Reward Function Parameters* | |
| $\alpha$ (exponential bonus) | 0.5 |
| $\beta$ (repetition penalty) | 0.9 |

### A.2 REWARD CALCULATION DETAILS

**Reasoning Format Reward Calculation.** The reasoning format reward $r_{\text{reasoning}}$ ensures proper output structure through binary verification of two components.

First, we check whether the reasoning process is properly enclosed within `<think>...</think>` tags. Second, we verify that the final answer is enclosed within `<answer>...</answer>` tags and contains a valid JSON format with required fields ("bbox_2d" and "point_2d" for segmentation tasks, "count" and "points" for counting tasks).

The reasoning format reward is defined as:

$$r_{\text{reasoning}} = 0.2 \cdot \mathbb{1}[\text{valid\_think}] + 0.8 \cdot \mathbb{1}[\text{valid\_answer}] \tag{5}$$

where $\mathbb{1}[\cdot]$ returns 1 if the format is valid and 0 otherwise. We assign higher weight to the answer format to emphasize producing correctly structured final outputs.

**Segmentation Task Accuracy Reward Calculation.** For referring segmentation tasks, we follow the practice of Seg-Zero (Liu et al., 2025a). The answer accuracy reward $r_{\text{answer}}^{\text{seg}}$ evaluates three components of the predicted output:

$$r_{\text{answer}}^{\text{seg}} = \frac{1}{3}(r_{\text{bbox}} + r_{\text{corner}} + r_{\text{point}}) \tag{6}$$

where each component is defined as:

$$r_{\text{bbox}} = \mathbb{1}[\text{IoU}(\text{bbox}_{\text{pred}}, \text{bbox}_{\text{gt}}) > 0.5] \tag{7}$$

$$r_{\text{corner}} = \mathbb{1}[\frac{1}{4}\sum_{i=1}^{4} d(\text{corner}_i^{\text{pred}}, \text{corner}_i^{\text{gt}}) < 10] \tag{8}$$

$$r_{\text{point}} = \mathbb{1}[d(\text{point}_{\text{pred}}, \text{point}_{\text{gt}}) < 30] \tag{9}$$

Here, $d(\cdot, \cdot)$ denotes Euclidean distance in pixels, and $\mathbb{1}[\cdot]$ is the indicator function.

**Counting Task Accuracy Reward Calculation.** For counting tasks, the answer accuracy reward $r_{\text{answer}}^{\text{count}}$ combines count accuracy with localization precision:

$$r_{\text{answer}}^{\text{count}} = \frac{1}{2}(r_{\text{count}} + r_{\text{localization}}) \tag{10}$$

where:

$$r_{\text{count}} = \mathbb{1}[\text{count}_{\text{pred}} = \text{count}_{\text{gt}}] \tag{11}$$

$$r_{\text{localization}} = \frac{|\{p_i : \min_j d(p_i, g_j) < 50\}|}{\max(|\mathcal{P}|, |\mathcal{G}|)} \tag{12}$$

Here, $\mathcal{P} = \{p_1, \ldots, p_{|\mathcal{P}|}\}$ represents the predicted 2D points, $\mathcal{G} = \{g_1, \ldots, g_{|\mathcal{G}|}\}$ represents the ground truth 2D points, and the localization reward measures the fraction of correctly localized points with respect to the maximum of predicted and ground truth counts.

### A.3 LEARNING PROGRESSION ANALYSIS.

Figure 4 tracks performance on both in-domain and cross-task benchmarks throughout training. Notably, cross-task performance improves steadily alongside in-domain performance, indicating that the model is learning generalizable skills rather than overfitting to task-specific patterns.

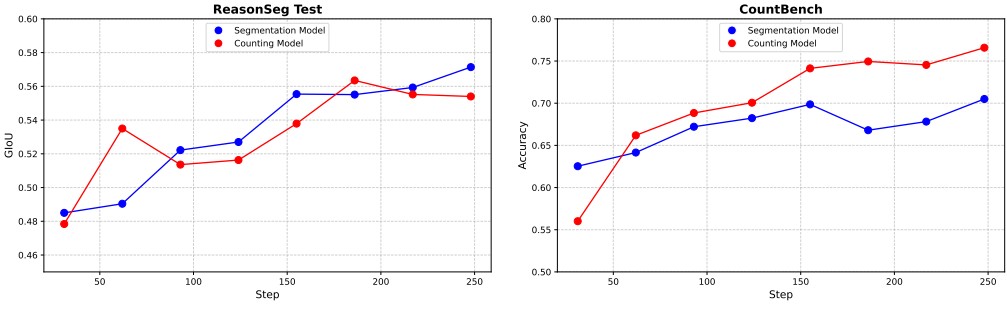

(a) Learning curves for RefCOCOg-test.  (b) Learning curves for CountBench.

Figure 4: Performance progression of segmentation-trained and counting-trained model on Reason-Seg test (a) and CountBench (b) during training. Both models show consistent improvement on both the in-domain and cross-task benchmarks.

### A.4 SUBTASK VALIDITY ANALYSIS

**Analysis Details.** We conducted an analysis of the model's performance across different subtasks to identify the validity and reliability of each subtasks. We utilized GPT-4o-mini (Hurst et al., 2024) to automatically evaluate the validity of subtask outputs, ensuring that executed subtasks are both accurate and relevant. We parsed the model's reasoning chains to extract individual subtask executions and their outputs, then prompted GPT-4o-mini (Hurst et al., 2024) with the given context of current image, question query, ground truth answer, and subtask output along with predicted outputs to determine validity. We also analyzed the reasoning chains of the VCoT (RL) baseline for comparison. As Table 7 shows, the model achieves high validity across most subtasks, demonstrating its reliability of subtask execution.

**GPT Evaluation Details.** Here we provide the exact prompt used to evaluate subtask validity with GPT-4o-mini:

Table 7: Validity of executed subtasks during inference. Abbreviations: q.d. - query_decomposition, o.d. - object_detection, p.s. - pixel_segmentation, a.r. - attribute_recognition, c.p. - captioning, r.r. - relation_reasoning, o.c. - object_counting, v.f. - verification, s.m. - summary. Both models are trained on RefCOCOg and evaluated on CountBench.

| Evaluation Model | Subtasks | | | | | | | | | Avg. | VCoT (RL) |
| | q.d. | o.d. | p.s. | a.r. | c.p. | r.r. | o.c. | v.f. | s.m. | | Sentences |
|---|---|---|---|---|---|---|---|---|---|---|---|
| GPT-4o-mini | 100.0 | 77.0 | 77.7 | 58.2 | 64.3 | 59.6 | 77.4 | 67.7 | 74.0 | 68.4 | 63.4 |

---

**Subtask Validity Evaluation Prompt**

You are an expert evaluator for a Vision-Language Model's reasoning trace.
Given one image, the question, ground-truth answer, the model's final prediction, and the parsed reasoning steps (grouped by subtask tags), evaluate each step for factual correctness and usefulness.
Instructions:
- Judge each numbered step independently, but consider the global context (image + question + GT + final prediction).
- If a step includes a subtask output like `<object_detection>` or `<object_counting>` with JSON, verify the payload matches the image (e.g., counts, classes, attributes, locations) at a high level. Perfect pixel accuracy is not required; focus on semantic correctness.
- Label each step as "correct" or "incorrect" and provide a brief justification (max 20 words). If uncertain, choose the best judgment from the image.
Return a strict JSON object with this schema:
{{ "evaluation": [ {{ "step_index": `<int>`, "evaluation": "correct" — "incorrect", "justification": `<string>` }}, ... ] }}
Question: {question}
Ground Truth Answer: {gt}
Model Predicted Answer: {pred}
Parsed Reasoning Steps:
{steps_block}

---

For the VCoT (RL) baseline, we evaluate the validity of each sentence in the reasoning chain using the following prompt:

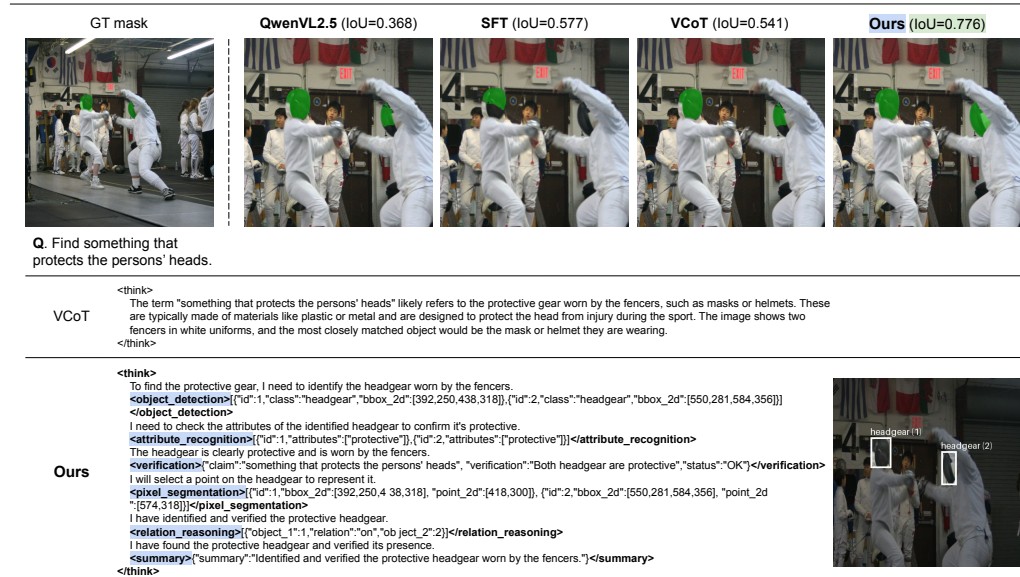

Figure 5: Qualitative example on the ReasonSeg benchmark. The model successfully identifies and localizes the referred object by composing multiple subtasks.

---

**VCoT (RL) Sentence Validity Evaluation Prompt**

You are an expert evaluator for a Vision-Language Model's reasoning trace.
Given one image, the question, ground-truth answer, the model's final prediction, and the parsed reasoning sentences (CoT without tags), evaluate each sentence for factual correctness and usefulness.
Instructions:
- Judge each numbered sentence independently, but consider the global context (image + question + GT + final prediction).
- For each sentence, decide if it's factually correct with respect to the image and whether it is relevant/helpful for solving the problem.
- Label each sentence as "correct" or "incorrect" and provide a brief justification (max 20 words). If uncertain, choose the best judgment from the image.
Return a strict JSON object with this schema:
{{ "evaluation": [ {{ "step_index": <int>, "evaluation": "correct" — "incorrect", "justification": <string> }}, ... ] }}
Question: {question}
Ground Truth Answer: {gt}
Model Predicted Answer: {pred}
Parsed Reasoning Steps:
{steps_block}

---

## A.5 ADDITIONAL QUALITATIVE RESULTS

**Qualitative Example on ReasonSeg Benchmark.** Figure 5 shows additional examples where the model successfully utilizes learned skills to solve complex reasoning tasks in the ReasonSeg benchmark.

**Qualitative Example on SomethingSomethingV2.** To assess generalization beyond our predefined subtask library, we evaluated on the SomethingSomethingV2 action recognition dataset, which requires temporal reasoning not explicitly covered in our training scope. Figure 6 shows a success case where the model correctly identifies "Opening something" by decomposing the video into ob-

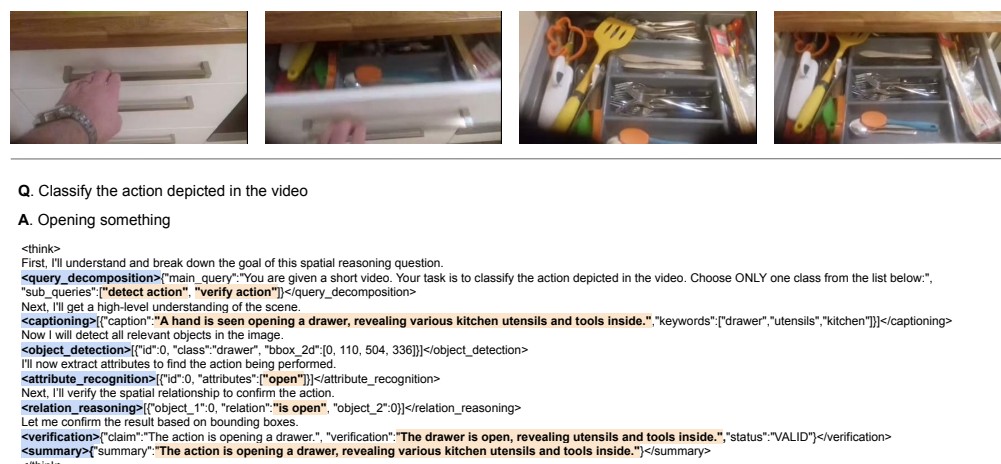

**Q**. Classify the action depicted in the video

**A**. Opening something

<think>
First, I'll understand and break down the goal of this spatial reasoning question.
**<query_decomposition>**{"main_query":"You are given a short video. Your task is to classify the action depicted in the video. Choose ONLY one class from the list below:", "sub_queries":[**"detect action"**, **"verify action"**]}</query_decomposition>
Next, I'll get a high-level understanding of the scene.
**<captioning>**[{"caption":**"A hand is seen opening a drawer, revealing various kitchen utensils and tools inside."**,"keywords":["drawer","utensils","kitchen"]}]</captioning>
Now I will detect all relevant objects in the image.
**<object_detection>**[{"id":0, "class":"drawer", "bbox_2d":[0, 110, 504, 336]}]</object_detection>
I'll now extract attributes to find the action being performed.
**<attribute_recognition>**[{"id":0, "attributes":[**"open"**]}]</attribute_recognition>
Next, I'll verify the spatial relationship to confirm the action.
**<relation_reasoning>**[{"object_1":0, "relation":**"is open"**, "object_2":0}]</relation_reasoning>
Let me confirm the result based on bounding boxes.
**<verification>**{"claim":"The action is opening a drawer.", "verification":**"The drawer is open, revealing utensils and tools inside."**,"status":"VALID"}</verification>
**<summary>**{"summary":**"The action is opening a drawer, revealing various kitchen utensils and tools inside."**}</summary>
</think>
<answer>{"answer":**"Opening something"**}</answer>

Figure 6: Qualitative example on the SomethingSomethingV2 Dataset.

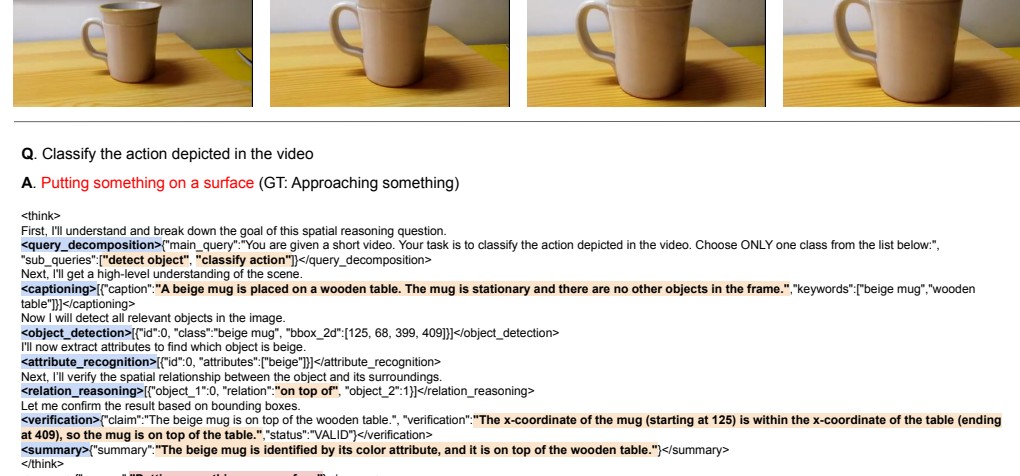

**Q**. Classify the action depicted in the video

**A**. Putting something on a surface (GT: Approaching something)

<think>
First, I'll understand and break down the goal of this spatial reasoning question.
**<query_decomposition>**{"main_query":"You are given a short video. Your task is to classify the action depicted in the video. Choose ONLY one class from the list below:", "sub_queries":[**"detect object"**, **"classify action"**]}</query_decomposition>
Next, I'll get a high-level understanding of the scene.
**<captioning>**[{"caption":**"A beige mug is placed on a wooden table. The mug is stationary and there are no other objects in the frame."**,"keywords":["beige mug","wooden table"]}]</captioning>
Now I will detect all relevant objects in the image.
**<object_detection>**[{"id":0, "class":"beige mug", "bbox_2d":[125, 68, 399, 409]}]</object_detection>
I'll now extract attributes to find which object is beige.
**<attribute_recognition>**[{"id":0, "attributes":["beige"]}]</attribute_recognition>
Next, I'll verify the spatial relationship between the object and its surroundings.
**<relation_reasoning>**[{"object_1":0, "relation":**"on top of"**, "object_2":1}]</relation_reasoning>
Let me confirm the result based on bounding boxes.
**<verification>**{"claim":"The beige mug is on top of the wooden table.", "verification":**"The x-coordinate of the mug (starting at 125) is within the x-coordinate of the table (ending at 409), so the mug is on top of the table."**,"status":"VALID"}</verification>
**<summary>**{"summary":**"The beige mug is identified by its color attribute, and it is on top of the wooden table."**}</summary>
</think>
<answer>{"answer":**"Putting something on a surface"**}</answer>

Figure 7: Failure case on the SomethingSomethingV2 Dataset.

ject detection, attribute recognition, and relation reasoning. This demonstrates that learned compositional strategies can extend to novel domains when the required operations align with available subtasks.

Figure 7 shows a failure case for "Approaching something," where the model applies spatial decomposition frame-by-frame but misses the temporal dynamics, predicting "Putting something on a surface" instead. While incorrect, the model generates coherent reasoning based on static spatial information, demonstrating structured behavior even on tasks requiring subtasks absent from our library.

## A.6 PROMPT TEMPLATES

Our prompt is designed to guide the model to initiate generating structured reasoning chains using predefined subtasks. We define a system instruction that outlines the task, available subtasks, and

specific rules and an example for formatting the output. To illustrate the expected behavior, we include a detailed example demonstrating the use of subtasks within the reasoning process.

---

**System Instruction**

You are a helpful assistant. Your task is to accurately find objects or answer questions based on the user's query.
To do this, you must reason step-by-step within `<think><\think>` tags. During your reasoning, you will use structured subtask tags to organize your visual analysis and reasoning process. Follow a logical workflow: first understand the query and the scene, then analyze the visual details, reason about the findings, and finally synthesize your answer.

**AVAILABLE SUBTASK TAGS** (Use these tools inside your `<think>` block whenever you need to perform a specific task):

{Subtasks}

{Rules}

{Example}

---

**User Prompt**

`<image>`
Question: "Please find "{Question}" with bboxs and points."

---

**Task-specific Rules**

**IMPORTANT RULES**:
- Your entire reasoning process MUST be enclosed in one `<think>...</think>` block.
- You MUST include explanatory text AND subtask tags within the `<think>` block.
- Each subtask tag's content MUST be a valid JSON object or array as specified, with NO extra text.
- Your final output, outside the `<think>` block, MUST be in an `<answer>...</answer>` tag and contain ONLY the JSON array of the final objects, formatted as [{"bbox_2d": [x1,y1,x2,y2], "point_2d": [x,y]}].

---

**Task-specific Example**

**EXAMPLE**:
Question: "Please find "'the yellow taxi'" with bbox and points."
`<think>`
Straightforward object detection task. Scanning for vehicles.
`<object_detection>` [{"id":1, "class":"taxi", "bbox_2d": [150,280,450,420]}, {"id":2, "class":"car", "bbox_2d": [500,300,700,440]}] `</object_detection>`
Checking color attributes to confirm it's yellow.
`<attribute_recognition>`[{"id":1, "attributes": ["color:yellow", "taxi_sign", "license_plate"]}] `</attribute\_recognition>`
Found it! Getting the center point.
`<pixel_segmentation>`[{"id":1, "bbox_2d": [150,280,450,420], "point_2d": [300,350]}] `</pixel_segmentation>`
`</think>`
`<answer>`[{"bbox_2d": [150,280,450,420], "point_2d": [300,350]}] `</answer>`

---

### A.7 Details on ReasonCount Benchmark

To evaluate our model's compositional reasoning capabilities more thoroughly, we introduce ReasonCount, a benchmark that complements existing simple counting benchmarks like CountBench with more complex counting tasks requiring multi-step reasoning and visual understanding. While CountBench focuses on basic object counting, ReasonCount tests whether models can generalize their learned subtasks to handle complex attribute-based counting and arithmetic reasoning.

**Dataset Construction.** ReasonCount is constructed by filtering and processing data from the PixMo-Point dataset (Deitke et al., 2025), which contains diverse real-world scenes with annotated objects.

For **complex attribute counting**, we select samples with long, complex referring expressions from the original dataset, ensuring questions require deeper reasoning about object attributes and relationships:

- Please count all 'dogs that are standing' in the image.
- Please count all 'photographs in nature' in the image.

For **arithmetic counting**, we create new questions that require adding or subtracting counts of different object categories, testing multi-step numerical reasoning:

- What is the total count of 'odd-numbered cards' and 'face cards'?
- What is the count of 'silverware' minus the count of 'knife'?

ReasonCount contains 1,000 questions of each type (2,000 total), providing a more challenging evaluation that tests both compositional subtask usage and cross-task generalization capabilities beyond what simple counting benchmarks can assess.

### A.8 Discussion on Limitations and Future Work

**Limitations.** A limitation of our current framework is its reliance on a pre-defined, hand-crafted library of subtasks. Consequently, the model's performance is bound by the expressiveness of this library, and its ability to generalize to tasks requiring fundamentally new primitive subtasks remains a challenge.

**Future directions.** The step-by-step subtask structure our model learns creates opportunities for smarter inference strategies. Rather than simply generating outputs greedily, we can treat the verifiable subtasks as decision points in a search process, enabling more sophisticated planning algorithms like Monte Carlo Tree Search or Tree of Thoughts (Yao et al., 2023; Long, 2023). This would let the model explore different ways to break down problems, eliminate poor choices early using our verifiable subtasks, and find better solutions—all without additional training.

### A.9 Learning Curves During RL Training

Here we provide learning curves showing the progression of reasoning format reward, answer accuracy reward and subtask format reward during RL training. As shown in the Figure 8, the reasoning format rewards converges quickly, indicating that the model learns to generate properly formatted reasoning chains early in training. Additionally, it can be seen that both the answer accuracy reward and the subtask format reward steadily increase together. This demonstrates that the multiplication of these reward components effectively guides the model to gain more reward by generating valid subtasks that lead to correct answers.

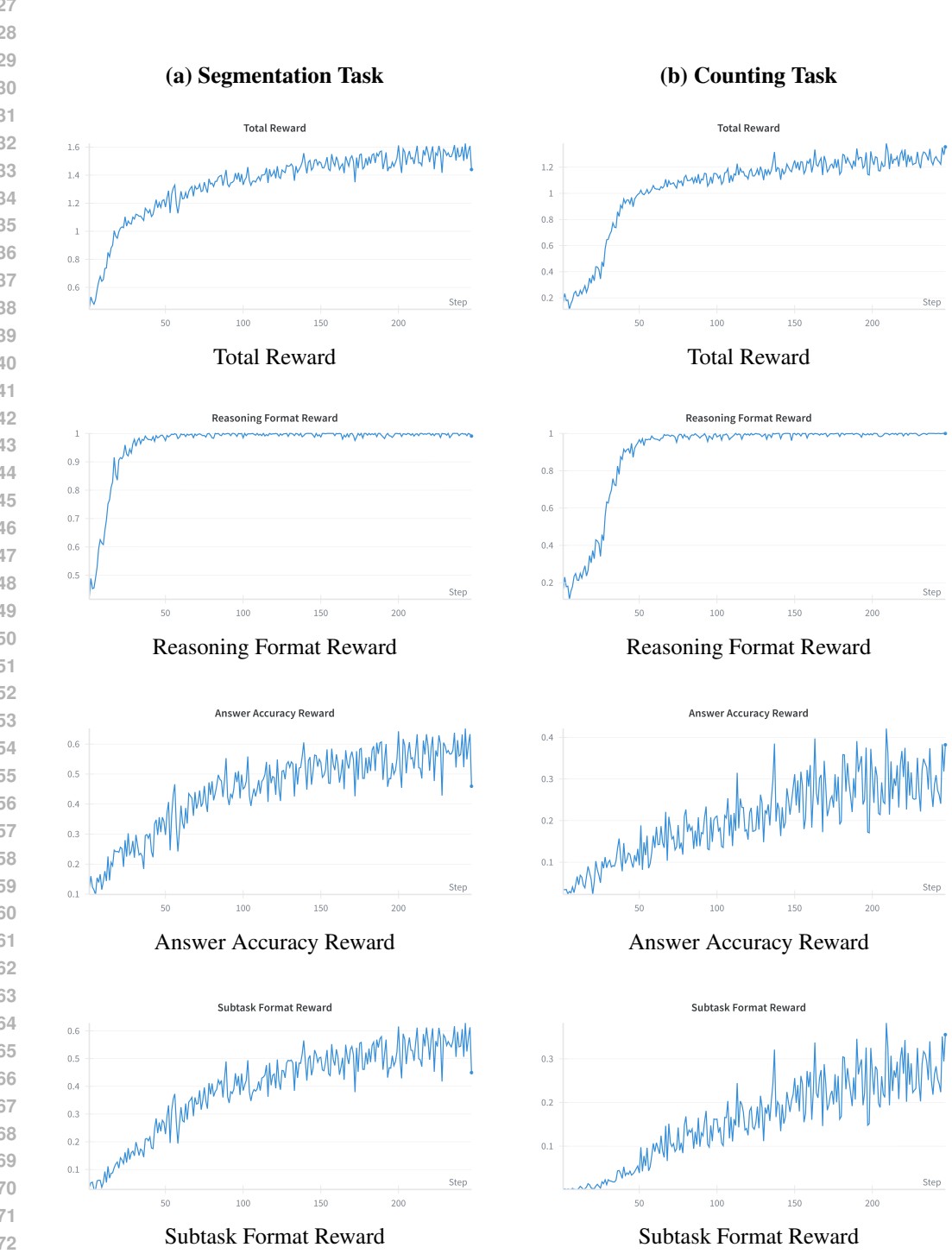

**(a) Segmentation Task**   **(b) Counting Task**

Total Reward

Total Reward

Reasoning Format Reward

Reasoning Format Reward

Answer Accuracy Reward

Answer Accuracy Reward

Subtask Format Reward

Subtask Format Reward

Figure 8: Reward learning curves during training on two different tasks. The left column shows results from the Segmentation task, and the right column shows results from the Counting task. Each row compares a different component of the reward. As shown in the figure, the reward components for both tasks exhibit a steady increase throughout training, indicating stable and effective learning.

