# OpenReview forum: "Reinforcement-Guided Subtask Decomposition for Unified Vision-Language Learning"
_ICLR.cc/2026/Conference — ICLR 2026 Conference Desk Rejected Submission_

### Official Review · Reviewer_8Q6b · 2025-10-29

**Soundness:** 3
**Presentation:** 3
**Contribution:** 2
**Rating:** 4
**Confidence:** 4

**Summary:**

The authors present a reinforcement learning framework to improve the cross-task generalization of Vision-Language Models (VLMs). The core idea is to train models to decompose problems into a sequence of verifiable subtasks. A key contribution is a composite reward function that provides a dense, format-based signal for reasoning steps, alleviating the need for step-by-step annotations. Experiments show the method achieves strong zero-shot performance and high efficiency using a 3B model trained on only 1k samples

**Strengths:**

1) Promising Approach: The paper addresses the critical problem of generalization in VLMs. The proposed compositional learning approach is a promising departure from standard fine-tuning and a valuable direction for building more robust and general models.

2) Practical Reward Design: The verifiable reward function is a clever solution to the sparse reward problem in multi-step reasoning. By using automated format verification, the framework provides dense guidance without expensive process supervision, making it highly practical.

3) Efficiency and Empirical Results: The method demonstrates a clear advantage in zero-shot generalization. Achieving this with a 3B model on only 1k samples highlights its remarkable data and computational efficiency, outperforming larger, specialized models.

**Weaknesses:**

1) The paper claims to learn "compositional, reusable fundamental skills," but evaluation is limited to dense vision benchmarks. Given the general nature of the subtask library, **it is unclear why more comprehensive benchmarks like MMStar or MMBench were not included.** Additionally, the reliance on a manually crafted subtask library may limit the model's ability to scale to tasks requiring entirely new atomic capabilities, potentially posing a limitation compared to other RL-related paradigms.

2) The reward function focuses on **the syntactic correctness of subtask outputs**, not their semantic factuality. This design could allow the model to learn incorrect reasoning paths that coincidentally produce a correct final answer, which might negatively impact generalization. **Why not incorporate intermediate outcome verification?** Many existing methods have provided dense, process-based rewards [1-2].

3) The paper lacks **scaling experiments for the proposed framework.** It is unclear why the 1k data sample size was chosen and how performance would change with more data (e.g., 10k samples, still far less than SFT). Similarly, how would the framework perform on a larger base model? A more capable model might better address the concerns in Weakness #2. This analysis is crucial for understanding the framework's full potential and robustness.

---

[1] Free Process Rewards without Process Labels

[2] Stepwise guided policy optimization:Coloring your incorrect reasoning in grpo.

**Questions:**

Please see the weakness section above.

---

> ### Author Response · Authors · 2025-11-18
> **Response to reviewer 8Q6b**
>
> We thank the reviewer for recognizing our approach as promising, highlighting our practical reward design, and noting our efficiency with a 3B model. We address below each weakness raised.
>
>
> ### **Weakness 1: Additional evaluations on general VQA and generalization**
> We did experiments on MMStar and MMBench. Our trained model achieves 55.62% on MMStar (pretrained Qwen2.5-VL-3B baseline: 55.87%) and 79.47% on MMBench (pretrained baseline: 77.06%). These results show that our method maintains or improves general VQA capabilities.
>
> Regarding the scalability concern, we argue that the challenge of handling entirely new capabilities applies to RL approaches, not unique to structured subtasks. Free-form RL methods like VCoT-RL also fail to discover new capabilities, as evidenced by severe cross-task degradation in Table 2 (-18.6 GIoU, -4.7% accuracy). The key difference is that structured guidance enables compositional transfer across related tasks, while unstructured methods learn task-specific patterns that fail to generalize.
>
> Our evaluation on Something-Something V2 demonstrates this advantage, as detailed in our response to Weakness 3 of reviewer ixam. The model successfully solves novel tasks by composing learned subtasks when applicable, as shown in the Figure 6 of the supplementary material. We acknowledge that extending our framework to incorporate automatic subtask discovery mechanisms, as discussed in our response to Weakness 5 of reviewer L3p4, would further broaden applicability while preserving these compositional learning benefits.
>
>
> ### **Weakness 2: Rationale for format-based verification**
> Our design choice reflects trade-offs between computational efficiency, flexibility, and scalability. Intermediate outcome verification requires judge models for each subtask type, significantly increasing computational cost. Additionally, when introducing new subtasks, judge models must be retrained or redesigned, and performance becomes bounded by judge model capability.
>
> Despite using format-based verification, our empirical analysis shows that subtask validity improves during training. As shown in Supplementary Table 7, when evaluated using GPT-4o-mini for semantic correctness, our method achieves 68.4% average subtask validity compared to 63.4% for VCoT-RL. This suggests that our structured format encourages semantically valid reasoning even without explicit semantic supervision. While semantic verification could provide additional benefits, our results demonstrate that format-based verification already enables effective learning of valid reasoning patterns.
>
> ### **Weakness 3: Model scale-up**
> Regarding model scaling to 7B, we are currently conducting experiments as discussed in Weakness 3 from Reviewer L3p4, and will provide results as they become available.
>
> Regarding data scaling beyond 1K samples, our experiments focused on demonstrating sample efficiency—that compositional learning enables effective generalization from limited data. As Table 5 shows, 1K samples already achieve competitive performance with larger specialist models. Exploring performance at 10K+ samples remains an interesting direction for future work.

---

> > ### Author Response · Authors · 2025-11-26
> > **Additional response to reviewer 8Q6b**
> >
> > We conducted experiments with reduced training data proportions to evaluate scaling behavior:
> >
> > | Data | ReasonSeg-val (GIoU) | ReasonSeg-test (GIoU) | CountBench (Acc) | Average |
> > |------|---------------|----------------|------------|---------|
> > | 25%  | 56.1         | 52.5          | 68.4      | 59.0   |
> > | 50%  | 59.3         | 56.0          | 72.7      | 62.7   |
> > | 100% | 59.4         | 57.1          | 73.3      | 63.3   |
> >
> > The consistent improvement from 25% to 100% demonstrates positive scaling behavior. This suggests that training with larger datasets would likely yield further performance gains. While our current experiments focus on the 1000-sample regime to demonstrate sample efficiency, exploring performance at 10K+ samples remains a promising direction for future work to fully characterize the scaling properties of our framework.

---

> > > ### Author Response · Authors · 2025-12-03
> > > **Additional response to reviewer 8Q6b**
> > >
> > > Following your suggestion to explore larger data size, we conducted additional experiments with 4K training samples (400% of our original size). While this remains smaller than typical SFT datasets, it allows us to investigate scaling trends:
> > >
> > > | Data | ReasonSeg-val (GIoU) | ReasonSeg-test (GIoU) | CountBench (Acc) | Average |
> > > |------|---------------|----------------|------------|---------|
> > > | 100% | 59.4         | **57.1**          | 73.3      | 63.3   |
> > > | 400% | **59.7**        | 56.1           | **78.6**      | **64.8**   |
> > >
> > > The results show meaningful performance improvements, with the average score increasing from 63.3 to 64.8. This demonstrates that our framework benefits from additional training data and suggests larger datasets can achieve further gains.

---

### Official Review · Reviewer_uMKS · 2025-11-03

**Soundness:** 3
**Presentation:** 3
**Contribution:** 2
**Rating:** 4
**Confidence:** 4

**Summary:**

The paper introduces a reinforcement learning framework aimed at improving cross-task generalization in vision-language models (VLMs). The authors argue that simply scaling up model size and data does not yield truly general-purpose models. Instead, they propose teaching models to solve tasks by composing reusable and verifiable subtasks, thereby enabling them to adapt to new problems requiring similar visual-linguistic skills in different combinations. The key innovation lies in a dense, format-based reward function that allows the model to learn structured reasoning chains of subtasks without explicit supervision of intermediate reasoning steps. This reward guides the model to produce valid, interpretable decompositions while also achieving accurate final results.

**Strengths:**

The article is well-structured and fluently written, enabling readers to easily comprehend its main points. Furthermore, the principle of “recomposing the same primitive subtasks in different sequences” is both coherent and insightful.

**Weaknesses:**

The methodology appears to be quite intuitive and straightforward. Consequently, for such work, the critical evaluation lies in empirically verifying its effectiveness.

Regarding the experimental setup, I would like to confirm that you employed a Zero RL approach, meaning no task-specific SFT was conducted before the reinforcement learning phase. The reward curves suggest that the model rapidly satisfied the format reward (presumably set to a value of 1, a detail I could not locate in the manuscript), while the accuracy rewards for other subtasks improved more gradually.

This observation raises a question about the necessity of complex reasoning for the proposed subtasks. Based on the visualization examples provided, these tasks seem to demand more general visual knowledge than the intricate reasoning required in domains like mathematics or formal logic.

This leads me to hypothesize that the primary performance gains may stem from the synergistic effects between the various visual tasks, rather than being a direct result of the RL process itself. I would welcome your perspective on this interpretation, as I am keen to understand if I have misinterpreted any aspect of your work.

**Questions:**

Since the author mentioned in the abstract that the 3B parameter is one of their advantages, I would like to know if there are more comparison with some larger models (such as Qwen2.5-VL-32B). How does their performance compare?

The comparison with the baseline shows that the method is beneficial (which is usually a necessity for most work), and it also has advantages when compared with other methods (SFT, RL) and expert models. Therefore, I would like to know the comparison of the 3B model proposed in this paper with larger models. I have seen the comparison with Qwen2.5-VL-7B, does it under the setting of training-free and zero-shot? How much benefit would SFT bring directly to this size?

I was wondering if you have plans to (or already) release the model or its training data. The reason is that for the community, having access to the model and data is often essential for a deeper assessment of the work’s merits.

I will adjust my opinion on this article based on the author's response.

---

> ### Author Response · Authors · 2025-11-18
> **Response to reviewer uMKS**
>
> We thank the reviewer for their careful assessment and for recognizing the coherence of our compositional approach. We address below each weakness and question raised.
>
> ### **[Weaknesses]**
> ### **Weakness 1-3: Clarification on the role of RL**
> To clarify our experimental setup, we indeed employed a zero-RL approach without task-specific SFT. Your observation about the reward dynamics aligns with our design intent. The format reward saturates rapidly because it serves as a simple structural constraint, while the accuracy and subtask format rewards improve gradually as the model learns to execute vision-specific subtasks correctly.
> Your hypothesis about visual task synergies captures an important aspect, though we would frame it differently. We designed our framework specifically to enable models to discover and leverage these synergies through compositional learning. The key is that these synergies do not emerge automatically. Discovering which subtask compositions generalize across tasks requires exploration. This is precisely why we combine structured subtasks with RL. The structured subtasks define a compositional search space, while RL enables exploration within that space to discover generalizable strategies. The performance gains stem from this deliberate design, which enables learning reusable compositions rather than task-specific patterns.
>
> ### **[Questions]**
> ### **Question 1: Bigger baselines comparison**
> We have evaluated Qwen2.5-VL-32B as an additional baseline, which achieves 65.4 GIoU on ReasonSeg validation compared to our 3B model's 59.4 GIoU. This performance difference is expected given the substantial capacity gap. Our emphasis on 3B refers to achieving competitive performance with efficient model size, as demonstrated by our 3B model outperforming or matching 7B baselines across most benchmarks.
>
> ### **Question 2: Bigger case pre-trained model**
> Regarding the Table 5 evaluation settings, the Qwen2.5-VL-7B results represent zero-shot, training-free evaluation, while specialist models were trained on task-specific data (often exceeding 10K samples). Our 3B models were trained on only 1K samples per task.
> Regarding 7B SFT comparison, we have not yet trained 7B models with either SFT or our method due to resource constraints. However, we note that our 3B model already outperforms the 7B zero-shot baseline on most benchmarks, and the performance gap between our method and baselines remains consistent across experiments. We are conducting 7B scale-up experiments as discussed in our response to Weakness3 from Reviewer 8Q6b, which will include SFT comparisons.
>
> ### **Question 3: Code and model release**
> We are currently preparing the complete training code, evaluation scripts for benchmarks, model weights for trained models. All materials will be released publicly upon acceptance.

---

> ### Comment · Reviewer_uMKS · 2025-11-26
> **Reply to Authors**
>
> Thanks for your response.
>
> Beyond that, a 3B model equipped with your methods and RL exceeds a zero-shot 7B model. Can it also surpass a 7B model that is directly SFT on task-specific data?
>
> Some small-sized model works claim that their methods, when applied to smaller models, can outperform larger models that are simply scaled up with more data. I believe achieving this is the key point.

---

> > ### Author Response · Authors · 2025-11-27
> >
> > Thank you for this insightful question about comparing our 3B model with larger 7B baselines.
> >
> > It is natural that the 7B model has enhanced generalization performance in various benchmarks due to its high capacity; it still has a limitation in cross-task transferability when target tasks are not adjacent to the tasks used for training.
> >
> > The following table shows that our 3B model outperforms (or exhibits competitive performance to) the 7B SFT models on EgoOrientBench and V*, which are not closely related to RefCOCOg or PixMo-Point. These results imply the benefit of our approach.
> >
> > | Model Size | Training Data | ReasonSeg-val | ReasonSeg-test | CountBench | EgoOrientBench (Choice-Acc) | V* (Acc) |
> > |------|---------------|----------------|------------|--------------|----------------|------------|
> > | 7B  | - (zero-shot)         |   56.9    |   52.1    |   76.0    |     37.6      |   70.2    |
> > | 7B  | RefCOCOg (Segmentation)        |   63.1    |   58.4    |   84.9    |   37.6          | 71.2      |
> > | 7B | PixMo-Point (Counting)        |   37.1    |   30.1    |   85.5    | 35.4          | 71.7      |
> > | 3B  | - (zero-shot)         |   54.5    |   47.6    |   72.5    |     28.2      |   70.7    |
> > | Ours (3B)  | RefCOCOg (Segmentation)         |   59.5    |   57.1    |   73.3    |   36.8          | 72.8      |
> > | Ours (3B)  | PixMo-Point (Counting)          |   56.9   |   55.4    |   76.6    | 37.8          |73.3      |

---

### Official Review · Reviewer_ixam · 2025-11-05

**Soundness:** 3
**Presentation:** 3
**Contribution:** 3
**Rating:** 8
**Confidence:** 3

**Summary:**

This paper presents a framework for enabling vision-language models (VLMs) to generalize across structurally different tasks by learning to decompose a query into a chain of verifiable subtasks and then execute those subtasks to produce an answer. Instead of requiring annotated reasoning chains, the model is trained via a reinforcement-learning method (Group Relative Policy Optimization, GRPO) and a novel reward that combines: (i) format-based verification of generated subtasks and (ii) final task-accuracy. It uses a predefined library of primitive subtasks (object detection, segmentation, attribute recognition, counting, captioning etc) each with a strict JSON output format. After training on one or two tasks (e.g., segmentation task RefCOCOg or counting task PixMo-Point), the resulting 3B-parameter model is evaluated zero-shot on multiple other tasks (ReasonSeg, CountBench, ReasonCount, EgoOrientBench, V*Bench) and shows superior cross-task generalization compared to supervised fine-tuning (SFT) or unstructured reasoning (Visual Chain-of-Thought + RL). The central claim is that composition of reusable subtasks is a more efficient and generalizable path to broad vision-language understanding than monolithic task-specific fine-tuning.

**Strengths:**

Novel reward design enabling dense feedback for reasoning chains. : The paper’s reward r = r_reasoning + r_subtask·r_answer combines structural verification of intermediate subtask formats and final performance. This allows training without annotated reasoning chains yet still learning structured decomposition.

Predefined subtask library with explicit JSON schema for auto-verification : Table 1 lists subtasks (query decomposition, object detection, pixel segmentation, attribute recognition, captioning, relation reasoning, object counting, verification, summary) each with strict output format, enabling automatic verification and hence dense feedback.

Strong cross-task generalization with minimal supervised data.: The method trains on only ~1,000 samples from RefCOCOg or PixMo-Point, yet shows improvements in both in-domain and many unseen tasks (segmentation, counting, spatial VQA) compared to baselines. Figures 2 and 3 show the model’s generated subtask sequences (e.g., object_detection → object_counting → captioning → verification) and provide insight into how the learned policy composes primitives rather than memorizing end-to-end.

Clear ablation studies of reward components: Table 3 and Table 4 isolate the effect of the exploration bonus and repetition penalty in r_subtask, showing their importance (e.g., exploration bonus + repetition penalty gives best GIoU of 59.5).

**Weaknesses:**

Predefined subtask library limits flexibility and may bake in bias.: The system relies on a manually defined list of subtasks and strict schemas (Table 1). While this allows verification, it may constrain the model to the authors’ chosen primitives and limit adaptation to tasks requiring new primitives.

Dependence on fairly strong pretrained modules / frozen decoders.: For segmentation tasks, the paper uses a frozen SAM2-Large decoder, and the base policy is Qwen2.5-VL-3B. This means that the “learning” is constrained by the underlying models and may hide where improvement comes from (i.e., large backbone rather than subtask decomposition). This complicates isolation of the contribution.

Limited evaluation on truly novel tasks outside the subtask library’s design; risk of overfitting to the library.: While the cross-task tasks are structurally different, they still map well to the chosen primitive set (counting, segmentation, spatial VQA). It would be stronger to evaluate on tasks requiring new subtask types (e.g., reasoning about temporal sequences, action recognition) to test how well the model generalizes beyond library coverage.

Compute and training cost / practicality not deeply discussed.: The paper uses GRPO with group sampling (G outputs per input) and fine-tunes a 3B-parameter model with RL. The samples per input, number of updates, and wall-clock cost are only briefly described (batch size 16, 8 samples). It would help to see a compute-vs-gain trade-off, especially since RL on large models is expensive.

**Questions:**

How do you select the subtask library? What criteria determine which primitives are included, and how sensitive is performance to this choice?

What happens when a downstream task requires a subtask not in the library (e.g., action-recognition, temporal reasoning)? Does the model degrade gracefully?

Can you provide statistics on reasoning chain lengths and which subtasks are most frequently used/mis-used?

How sensitive is performance to hyperparameters α (exploration bonus) and β (repetition penalty)? You show some ablation but can you comment on general tuning strategy?
Have you observed failure cases where the model generates many useless subtasks (long chain) but still answers incorrectly? How do you detect/handle those?

---

> ### Author Response · Authors · 2025-11-18
> **[1/2] Response to reviewer ixam**
>
> We sincerely thank the reviewer for recognizing our novel reward design, strong cross-task generalization with minimal data, and clear ablations. We address below each weakness and question raised.
>
> ### **Weakness 1 & Question 1: Subtask library design and future extension**
> Our subtask library was developed through analysis of common vision-language operations combined with empirical observation of recurring patterns in existing VLM outputs. We deliberately limited our scope to spatial and compositional perception tasks to demonstrate the feasibility of learning reusable subtasks and their cross-task transfer, rather than attempting to cover all possible reasoning types.
>
> While we designed our library to cover common visual reasoning operations, we recognize that extending this framework to a more general and automatically discoverable subtask library is crucial for broader applicability for future research.
>
> ### **Weakness 2: Isolating method contribution from pretrained components**
> We would like to clarify how our contribution is distinct from the underlying model capacity. Our method demonstrates consistent improvements across both settings (Table 2), showing that the benefits arise from compositional subtask learning rather than decoder capacity.
> The cross-task transfer results provide particularly strong evidence—models trained on counting (without SAM2) successfully transfer to segmentation tasks, and vice versa. Since all architectural components remain constant across methods, this improvement can only stem from learning reusable compositional skills during training. Furthermore, our 3B model achieves competitive or superior performance compared to much larger specialized models (Table 5), demonstrating that compositional learning provides efficiency gains independent of backbone capacity.
>
> ### **Weakness 3 & Question 2: Generalization beyond subtask library coverage**
> We have evaluated on action recognition using Something-Something V2, a video understanding task requiring reasoning beyond our training scope. Qualitative examples are provided in the Figure 6 and 7 of the supplementary material.
>
> Our results demonstrate that vision-centric subtasks can extend to novel domains. The model successfully recognizes certain actions through compositional spatial analysis. For instance, it correctly identifies "Opening something" by decomposing the video into object detection, attribute recognition, and spatial relation reasoning. This shows that learned compositional strategies apply beyond the original training tasks when the underlying visual operations remain relevant.
>
> For actions requiring explicit temporal dynamics, such as "Approaching something," the model attempts spatial decomposition but produces incorrect predictions. However, the model still generates structured, relevant reasoning chains (e.g., analyzing object-surface relations to predict "Putting something on a surface").
>
> ### **Weakness 4: Computational cost analysis**
> Our training configuration uses batch size 16, G=8 samples per input, and takes approximately 8 hours on 8×A6000 GPUs (3B model, 1K samples, 4 epochs). Compared to baselines, our training cost is comparable to VCoT-RL (both use GRPO) with single training step times of 101.94s (Ours) vs 100.04s (VCoT-RL) vs 8.23s (SFT).
>
> For compute-vs-gain trade-off, our method achieves +0.8 GIoU improvement on ReasonSeg-val over VCoT-RL with comparable training cost. Compared to SFT, the additional cost enables effective cross-task transfer: +0.8 accuracy on CountBench versus SFT's -0.2 degradation. We believe this computational cost is justified given the fundamental capability improvement in cross-task generalization.

---

> > ### Author Response · Authors · 2025-11-18
> > **[2/2] Response to reviewer ixam**
> >
> > ### **Question 3: Statistics on subtask usages**
> > On average, our model used 517 tokens in the response including reasoning and answer, whereas the VCoT model used 184 tokens. The fact that our model consumes more tokens is natural, as its reasoning—previously done purely through free-form text—now includes the execution of various subtasks, which naturally lengthens the chain.
> > The most frequently used subtasks were ``<object_detection>`` (35.7%) in the segmentation task and ``<object_counting>`` (50.5%) in the counting task. This indicates that the model effectively selects subtasks that are appropriate for the given task.
> >  The most incorrectly used subtask, based on validity checks performed using GPT-4o-mini, was ``<attribute recognition>`` (58.2%), while we believe this lower validity arises from the inherently larger set of attributes (color, size, etc.) that need to be considered compared to other subtasks.
> >
> > ### **Question 4: Additional study on hyperparameters**
> > Here we provide both empirical sensitivity analysis and insights into our tuning strategy.
> >
> > Regarding hyperparameter sensitivity, Table 4 in the main paper shows ablation results for β (repetition penalty). We provide additional ablation results for α (exploration bonus) under the same experimental setup (trained on RefCOCOg, evaluated on ReasonSeg-val and CountBench):
> >
> >
> > | α (exploration bonus) | ReasonSeg-val (GIoU) | CountBench (Acc) |
> > |-----------------------|---------------------|------------------|
> > | 0.3                   | 56.9                | 64.5             |
> > | 0.5                   | **59.4**                | **73.2**             |
> > | 0.7                   | 53.3                | 67.0             |
> >
> >
> > As described in Section 3.3, α encourages diverse subtask usage. The results demonstrate that α=0.5 achieves the highest performance, with α=0.3 leading to insufficient decomposition and α=0.7 producing overly long chains that hurt performance.
> >
> > Our tuning strategy was guided by observed failure modes during development. We initially encountered models that either used minimal decomposition or exhibited excessive repetition of the same subtask without making progress. The exploration bonus α addresses the former by rewarding diverse subtask usage, while the repetition penalty β mitigates the latter.
> >
> > Regarding failure cases with long but unhelpful chains, our reward design handles these naturally through its multiplicative structure. Since the final reward is $r_{subtask}\times r_{answer}$, any chain that produces an incorrect answer receives zero reward regardless of subtask quality. This provides clear negative feedback that prevents the model from generating elaborate but incorrect reasoning.

---

### Official Review · Reviewer_L3p4 · 2025-11-07

**Soundness:** 2
**Presentation:** 3
**Contribution:** 3
**Rating:** 4
**Confidence:** 3

**Summary:**

The authors propose a reinforcement learning (RL) framework for vision-language models (VLMs) in which, instead of producing free-text chains of thought (CoT), the model composes a sequence of predefined, reusable primitive subtasks. The key hypothesis is that enforcing this compositional prior within the CoT enables better cross-task generalization -- i.e., a model trained on one task can adapt to new tasks by reusing learned primitives.

The method is trained using GRPO with a composite reward comprising reasoning-format, subtask-format, and answer-accuracy components. Experiments are conducted by training on one of two tasks—segmentation (RefCOCOg) or counting-by-localization (PixMo-Point)—and evaluating across segmentation (ReasonSeg), counting (CountBench and the proposed ReasonCount), and spatial VQA (EgoOrientBench, V*Bench).

Baselines include (i) a zero-shot VLM, (ii) supervised fine-tuning (SFT) on correct final answers, and (iii) free-text visual CoT with RL on the same data. Results show that the proposed method achieves superior cross-task performance and demonstrates effective reuse of primitive skills across higher-level tasks. The paper also provides ablation studies and qualitative analyses to support these findings.

**Strengths:**

1. The proposed method to enable the model to learn to compose and reuse primitive visual operations (such as object detection, attribute recognition, etc.) through RL is adequately motivated. It is conceptually aligned to past works such as Neural Module Networks (non-RL) and ViperGPT (zero-shot only), but explores this in a CoT RL formulation.

2. The experiments with Qwen2.5-VL-3B model show better cross-task results compared to traditional free-text CoT-RL tuned model and SFT baseline. Further ablations show impact of introducing exploration bonus and repetition penalty in addition to traditional format and final-task reward.

3. Paper is largely well written with primary experiment details and results clearly described. Additional analysis on subtask validity and qualitative analysis is also informative.

**Weaknesses:**

1.  **Evaluation on multi-task training setting not reported**:  The experiments currently only evaluate cross-task performance, where the model is trained on individual task data (e.g., segmentation) and tested across tasks. In a more realistic scenario, training data for multiple tasks is available. The authors should thus also report performance for both the baselines and their method under **multi-task training** (e.g., jointly training on counting and segmentation). Without this result, it remains unclear whether, in a multi-task more realistic training scenario, standard free-text CoT RL tuning might already achieve comparable cross-task performance without requiring predefined primitive operations.

2. **Evaluation on compositional multi-step reasoning datasets**: Related to weakness 1, authors could consider reporting results on compositional multi-step VQA reasoning datasets such as GQA [1], where sequential application and reuse of operations are more likely to be beneficial and not trivially learned through standard free-text CoTs even in a multi-task training scenario.


3. **Model scale dependence**:  Results are only reported on the Qwen-2.5-VL-3B model. Resources permitting, it would strengthen the paper to show whether the findings hold for a larger model (e.g., Qwen-2.5-VL-7B). Larger models might exhibit more expressive and effective free-text CoTs through standard RL finetuning, and thereby potentially have less improvements from predefined primitive operations.

4. **Sample efficiency comparison not analyzed**:  A potential benefit of composing predefined primitive operations is improved sample efficiency (i.e. less training data might be required as primitive operations can be reused and learned across samples). The authors could report performance with varying proportions of training data (e.g., 10%, 25%, 50%, 100%) to highlight this potential advantage and more strongly motivate benefits of their approach since RL training data may not be easily available across tasks.


Relatively minor:

5. **Prompting strategy for free-text CoT model:** Authors can also have a baseline where model is just prompted to consider performing low-level relevant tasks such as object detection, attribute recognition, etc in it's CoT instead of a specific predefined format constrained to predefined operations.

6. **Claim of denser-reward:** The composite RL reward function introduces additional modulation terms such as an exploration bonus and a repetition penalty. While these are useful and effectively modulate the final task reward (as shown in the ablations), it is inaccurate to describe them as making the reward function denser. A denser reward is typically understood as a process-level reward that provides intermediate correctness feedback for subtasks in addition to the final task reward. This is not the case here, and the authors should consider rephrasing this claim.

[1] Hudson, Drew A., and Christopher D. Manning. "Gqa: A new dataset for real-world visual reasoning and compositional question answering." Proceedings of the IEEE/CVF conference on computer vision and pattern recognition. 2019.

**Questions:**

Please see weaknesses section.

---

> ### Author Response · Authors · 2025-11-18
> **Response to reviewer L3P4**
>
> We thank the reviewer for recognizing the motivation of our approach and its strong cross-task generalization results. We address below each weakness and question raised.
>
> ### **Weakness 1: Evaluation on multi-task training setting not reported**
> As you noted, multi-task training is indeed a practical and meaningful scenario. That said, we would like to clarify the core challenge inherent in our framework. As our methodology depends on task-specific answer rewards (e.g., IoU for segmentation, accuracy for counting), straightforward multi-task training is not immediately applicable without additional innovations in architecture or reward design. Nonetheless, we agree that this represents a highly promising direction for future research.
>
> ### **Weakness 2: Evaluation on compositional multi-step reasoning datasets**
> We have evaluated on GQA, achieving 59.91% (pretrained Qwen2.5vl-3B baseline 59.98%). We note that our evaluation already includes several compositional multi-step reasoning benchmarks. ReasonSeg requires decomposing complex referring expressions, V*Bench involves multi-step guided visual search, and ReasonCount tests compositional arithmetic. On these benchmarks, our method demonstrates substantial improvements (Table 2), validating the benefits of sequential operation reuse.
>
>
> ### **Weakness 3: Model scale dependence**
> We are currently conducting Qwen2.5-VL-7B experiments, though full optimization requires substantial computational resources and may extend beyond the rebuttal timeline. We will include any available results in the camera-ready version.
> We understand the concern that larger models might produce sufficiently expressive free-text CoTs to reduce our advantage. However, we believe the key issue is cross-task generalization, not in-domain reasoning quality. Larger models may generate better free-text CoTs for individual tasks, but will likely face the same cross-task transfer challenges without explicit compositional structure. Our structured subtasks force learning reusable primitives rather than task-specific patterns, which is orthogonal to model capacity. We expect 7B models to amplify both methods' in-domain performance while maintaining our cross-task advantage.
>
> ### **Weakness 4: Sample efficiency comparison not analyzed**
> We are currently conducting experiments with varying data proportions to systematically evaluate sample efficiency, and will update this response as results become available.
> We note that our baseline configuration already demonstrates strong data efficiency. Using only 1,000 samples per task, which is a small fraction of typical fine-tuning datasets (often 10K+ samples), our method achieves competitive or superior performance compared to much larger specialist models (Table 5). This suggests that compositional subtask learning enables effective learning from limited data, though systematic ablation across different data scales will provide more rigorous validation of this claim.
>
> ### **Weakness 5: Prompting strategy for free-text CoT model**
> We understand what you propose as prompting the model to consider simple subtasks during reasoning. We conducted this experiment during paper preparation and achieved 55.3 GIoU on the ReasonSeg validation set, which shows a slight improvement over the original Qwen2.5-VL-3B score of 54.5. However, our analysis of the reasoning process revealed that the model mostly relied on free-form text reasoning, largely ignoring the provided subtasks. This made it unsuitable as a meaningful baseline for evaluating compositional subtask learning, as the model was not actually executing the suggested subtasks in a verifiable way.
>
> ### **Weakness 6: Claim of denser-reward**
> Our reward operates at the output level rather than providing token-level process supervision. When we described our reward as "dense," we intended to convey that it provides feedback on multiple intermediate subtask outputs within each reasoning chain, rather than evaluating only the final answer. This contrasts with purely outcome-based approaches but differs from process-level rewards that can provide token-by-token correctness feedback. We note that reviewers ixam and 8Q6b recognized the value of this intermediate feedback structure as a strength. We will clarify this distinction in the manuscript to avoid ambiguity about the nature of our reward signal.

---

> > ### Author Response · Authors · 2025-11-26
> > **Additional response to reviewer L3P4**
> >
> > We conducted experiments with varying training data proportions to evaluate sample efficiency. The table below shows performance across different data scales.
> >
> > | Data | ReasonSeg-val (GIoU) | ReasonSeg-test (GIoU) | CountBench (Acc) | Average |
> > |------|---------------|----------------|------------|---------|
> > | 25%  | 56.1         | 52.5          | 68.4      | 59.0   |
> > | 50%  | 59.3         | 56.0          | 72.7      | 62.7   |
> > | 100% | 59.4         | 57.1          | 73.3      | 63.3   |
> >
> > With only 250 training samples, our method achieves 52.5 GIoU on ReasonSeg-test and 68.4% on CountBench. These results surpass the SFT baseline at 47.9 GIoU and the VCoT-RL baseline at 67.8%, both trained on 1000 samples. This demonstrates that our method enables effective cross-task generalization from limited training data. Performance improves consistently as data increases from 25% to 100%, with average scores rising from 59.0 to 63.3.
> >
> > These results validate both the sample efficiency of our approach and its positive scaling behavior. We note that our base training size of 1000 samples is already small, further highlighting the data efficiency of our framework.
> > We will include these results in the revised manuscript.

---

> ### Comment · Reviewer_L3p4 · 2025-11-27
>
> Thank you for the clarifications and updated results. The sample efficiency results are useful to show method's benefit on potentially data-scarce scenarios or tasks. Noted regarding ongoing experiments on Qwen2.5-VL-7B.
>
> A remaining concern:
>
> - For weakness 1 on multi-task/joint training, is it not possible to just have alternate tasks per batch (so simply for a given batch have the same model weights trained with reward computed either from only segmentation or counting )? Alternatively, a composite normalized-reward could also be done (basically scale IOU score b/w 0-1 and counting acc. b/w 0-1). I am unsure why this would require further modifications. If this is not possible, it should be noted as a limitation in the paper, since in realistic scenarios, one wants to be able to train on multiple tasks (and still show generalization over multiple evaluation tasks possibly not covered in training) rather than being constrained to a single training task.

---

> ### Author Response · Authors · 2025-11-27
>
> Thank you for the constructive suggestions. In our initial response, we may have overcomplicated the issue by suggesting design challenges. We agree that the approaches you outlined, such as alternating task batches or composite normalized rewards, could address the multi-task training scenario.
>
> Our focus on single-task training was primarily to demonstrate cross-task transfer in a setting where the model has no direct supervision on the target task, providing a sufficient proof-of-concept for our method.
> In multi-task training, the expanded scope of in-domain tasks would naturally improve performance on a broader range of evaluations; however, it also reduces the space of truly unseen tasks, making evaluation more difficult because the current benchmark lacks tasks that are sufficiently distant from those used for training.
> Note that, for our research objective of demonstrating cross-task generalization through subtask decomposition, the single-task setting provides a conceptually sufficient and appropriate testbed.

---

### Author Response · Authors · 2025-11-25

Dear Reviewers,

We hope our responses have helped clarify your concerns. As the discussion period ends next week, we wanted to gently follow up to see if there are any remaining questions or points you would like us to address further.

We remain happy to provide any additional clarifications. Thank you again for your valuable feedback and careful review.


Best regards,

Authors

---

### Author Response · Authors · 2025-12-03
**Summary of Rebuttal**

We sincerely thank all reviewers for their constructive and positive comments and we summarize the key discussions and additional experimental results presented during the rebuttal period.


---
We summarize our key contributions as follows:


* __Promising framework for cross-task generalization:__ We introduce a compositional RL framework that enables effective reuse of subtasks to tackle the challenge in VLM generalization, a direction acknowledged as highly promising by reviewers (__L3p4, uMKS, 8Q6b__).
* __Novel and practical reward design:__ We propose a novel verifiable reward mechanism that provides effective guidance for intermediate reasoning without expensive process supervision, a design noted by reviewers as a clever and practical solution to sparse reward problems (__ixam, 8Q6b__).
* __Strong cross-task performance:__ Our method demonstrates strong, superior cross-task performance (__ixam, L3p4__) over SFT and RL baselines, achieving these gains under remarkable data and parameter efficiency (__8Q6b__).


---
We present the summary of our responses to each reviewer as follows:


* __Semantic correctness without reasoning supervision:__ Our method effectively enables the model to learn valid reasoning patterns without additional computational cost from intermediate outcome verification. Our empirical analysis reveals that our novel verifiable reward encourages semantic correctness of the subtasks, achieving 68.4% subtask validity compared to 63.4% for the baseline (evaluated by GPT-4o-mini). This addresses the question regarding the reliability of intermediate reasoning from __Reviewer 8Q6b__.
* __Generalization beyond predefined subtask library:__ Qualitative results from Something-Something V2 (Supp. Figures 6-7) show our model successfully decomposes certain actions (e.g., "Opening something") into components from the subtask library, proving that our learned compositional strategies generalize well beyond the original training scope (__Reviewer ixam, 8Q6b__).
* __General VQA performance:__ On general VQA benchmarks (GQA, MMStar, MMBench), our model shows competitive performance to the pretrained baseline, demonstrating its robustness to comprehensive benchmarks (__Reviewer L3p4, 8Q6b__).
Sample efficiency: Only utilizing 25% of the original dataset size, our method shows competitive results compared to baselines trained on full datasets, addressing the inquiry of sample efficiency (__Reviewer L3p4__).
* __Data scalability:__ The performance of our method trends continues to improve as data scales to 400%, achieving an additional 1.5%p performance gain over benchmarks. This shows the capability and robustness of our method (__Reviewer 8Q6b__).
* __Benefits of compositional learning beyond model scaling:__ Despite the inherent capacity advantage of the 7B baseline in generalization, it still faces limitations when transferring to tasks that are not closely related to the training data. Our experiments show that the 3B model matches or outperforms the 7B SFT baseline on benchmarks such as EgoOrientBench and V*. This implies that the generalization capabilities learned through our compositional approach are distinct from the gains of model scaling (__Reviewer L3p4, 8Q6b, uMKS__).


---
These additional evaluations highlight the robustness of our framework. We are confident that our comprehensive rebuttal has sufficiently resolved the raised inquiries, despite the discussion phase was unfortunately interrupted before our responses could be taken into full consideration.
We sincerely appreciate your effort, and we hope that the above summary is helpful for your assessment.


Best regards,


Authors.

---

### Note · Program_Chairs · 2026-01-17
**Submission Desk Rejected by Program Chairs**

The following references in this submission do not refer to real documents and/or have major errors in bibliographic information:

     Liyuan Wang, Xingxing Zhang, Kuo Li, Liu Yang, and Yang You. A comprehensive survey on catastrophic forgetting in deep learning. IEEE Transactions on Neural Networks and Learning Systems, 2023.